# Antineoplastic Nature of *WWOX* in Glioblastoma Is Mainly a Consequence of Reduced Cell Viability and Invasion

**DOI:** 10.3390/biology12030465

**Published:** 2023-03-17

**Authors:** Żaneta Kałuzińska-Kołat, Katarzyna Kośla, Damian Kołat, Elżbieta Płuciennik, Andrzej K. Bednarek

**Affiliations:** 1Department of Molecular Carcinogenesis, Medical University of Lodz, 90-752 Lodz, Poland; katarzyna.kosla@umed.lodz.pl (K.K.); damian.kolat@umed.lodz.pl (D.K.); andrzej.bednarek@umed.lodz.pl (A.K.B.); 2Department of Functional Genomics, Medical University of Lodz, 90-752 Lodz, Poland; elzbieta.pluciennik@umed.lodz.pl

**Keywords:** WWOX, glioblastoma, GBM, viability, invasiveness, in vitro, in silico

## Abstract

**Simple Summary:**

*WWOX* encodes a protein whose deficiency severely impacts brain development. Therefore, the impact of this tumor suppressor on brain tumors should also be investigated, particularly so as its nature has scarcely been elucidated in regard to the most common malignant brain cancer, i.e., glioblastoma. Unlike most of the current research, this study took advantage of several glioblastoma cell lines and biological assays to determine the main processes by which *WWOX* exhibits anticancer properties in glioblastoma. In the majority of the included cellular models, *WWOX* was found to intensify apoptosis, suppress cell viability, diminish adhesion to the extracellular matrix, decrease the quantity of colonies, and reduce invasiveness. Some discrepancies in the impacts on tumor proliferation and growth were noted and these were explained using bioinformatics tools. Nevertheless, *WWOX* inhibited glioblastoma viability and invasiveness independently of the cellular model. This is valuable to the scientific community as it guides the pursuit of research directions that can entail the development of relevant anticancer approaches, eventually leading to solutions similar to the *WWOX* gene therapy that is now under investigation for its usefulness in *WWOX*-related epileptic encephalopathy.

**Abstract:**

Following the discovery of *WWOX*, research has moved in many directions, including the role of this putative tumor suppressor in the central nervous system and related diseases. The task of determining the nature of *WWOX* in glioblastoma (GBM) is still considered to be at the initial stage; however, the influence of this gene on the GBM malignant phenotype has already been reported. Because most of the available in vitro research does not consider several cellular GBM models or a wide range of investigated biological assays, the present study aimed to determine the main processes by which *WWOX* exhibits anticancer properties in GBM, while taking into account the phenotypic heterogeneity between cell lines. Ectopic *WWOX* overexpression was studied in T98G, DBTRG-05MG, U251MG, and U87MG cell lines that were compared with the use of assays investigating cell viability, proliferation, apoptosis, adhesion, clonogenicity, three-dimensional and anchorage-independent growth, and invasiveness. Observations presenting the antineoplastic properties of *WWOX* were consistent for T98G, U251MG, and U87MG. Increased proliferation and tumor growth were noted in *WWOX*-overexpressing DBTRG-05MG cells. A possible explanation for this, arrived at via bioinformatics tools, was linked to the TARDBP transcription factor and expression differences of *USP25* and *CPNE2* that regulate EGFR surface abundance. Collectively, and despite various cell line-specific circumstances, WWOX exhibits its anticancer nature mainly via a reduction of cell viability and invasiveness of glioblastoma.

## 1. Introduction

The WW domain-containing oxidoreductase (*WWOX*) gene was discovered by Bednarek et al. in the early 2000s and is associated with genomic instability as a consequence of its co-localization in the chromosomal common fragile site (CFS) denoted as FRA16D [1]. In the human genome, FRA16D is considered the second-most common chromosomal hotspot, outpaced only by the FRA3B site located within the *FHIT* gene [2,3,4]. Interestingly, *WWOX* contains one of the so-called human accelerated regions (HARs); hence it is possible to refer to this gene using the term “HAR6” [5,6]. The significance of *WWOX* laid the foundations for thorough investigations of its role not only in tumor suppression but also in regulating metabolism, cellular homeostasis, and DNA damage response [7,8,9,10,11,12,13,14]. *WWOX* gene expression was found to be downregulated or lost in several types of cancer [15] and this was related to the poor prognosis in tumors of the breast [16,17,18,19,20,21,22,23], liver [24,25,26], ovary [27,28], bladder [29,30], and kidney [31], as well as to glioblastoma [32,33,34].

Cancer promotion can occur through various processes, and numerous literature data indicate that reduction or loss of WWOX expression may regulate it by inhibiting programmed cell death and enhancing proliferation, migration, invasion, or genome instability [20,24,35]. A broad spectrum of WWOX action is possible owing to its ability to interact with other protein partners and thus modulate molecular mechanisms [36]. Most of these events appear to be cell-specific; thus, the anticancer WWOX activity in given cells would depend on its protein partners’ availability. The lack of such might explain the presence of tumor cells highly expressing WWOX [37]. Due to its genomic localization, WWOX tumor suppressor activity is much more frequently impaired by the loss of heterozygosity or homozygous deletion than by point mutations [37,38,39]. Initially recognized as a putative tumor suppressor in breast cancer, the *WWOX* gene is now in the spotlight for its possible role in, e.g., central nervous system (CNS) and related neurological disorders [2,9,32,40,41,42,43,44,45,46] such as *WWOX*-related epileptic encephalopathy (WOREE), for which a *WWOX* gene therapy was recently suggested as a proof-of-concept [47].

Due to the heterogeneous nature of glioblastoma (GBM), its treatment is a considerable challenge [48,49]; this neoplasm remains one of the most aggressive primary brain tumors. Considering the poor survival of affected oncological patients, a completely new approach may introduce effective opportunities for the targeting of cancer cells [50,51]. In addition to reduced life expectancy, its high recurrence rate only worsens the prognosis even after medical treatment [52,53] and is related to the intensive infiltration of healthy brain parts by the remaining tumor cells [54]. Available anti-GBM therapies have achieved only a marginal increase in the survival of patients survival due to difficulties such as the location of the tumor, the blood–brain barrier, and acquired resistance to radio- or chemotherapy [55,56]. Futile glioblastoma treatment solutions have prompted intensive research for a broader understanding of GBM progression-related molecular aberrations.

The task of determining the nature of *WWOX* in GBM is still considered to be at the initial stage [57], but the influence of this gene on the malignant phenotype of GBM has already been reported [33]. Because most of the available in vitro research does not consider multiple cellular models of GBM or a wide range of investigated biological assays, the present study aimed to determine the main processes by which *WWOX* exhibits anticancer properties in GBM, while taking into account the phenotypic heterogeneity between cell lines. In this study, we employed assays that investigate redox potential, proliferation, apoptosis, adhesion, clonogenicity, three-dimensional and anchorage-independent growth, gelatin degradation, and invasiveness. A possible explanation of discrepancies observed in one of the cell lines was provided using the bioinformatics tools.

## 2. Materials and Methods

### 2.1. Cell Lines and Culture Conditions

The glioblastoma cell lines T98G, DBTRG-05MG, U251MG, and U87MG were purchased from the European Collection of Cell Cultures (ECACC). T98G cells were grown according to the manufacturer’s protocol in Minimum Essential Media (MEM; Thermo Fisher Scientific, Naarden, The Netherlands) supplemented with 1% non-essential amino acids (NEAA; Gibco, Ghent, Belgium), DBTRG-05MG in RPMI 1640 supplemented with 1 mM sodium pyruvate (Gibco, Ghent, Belgium), whereas U251MG and U87MG in Eagle’s Minimum Essential Media (EMEM) supplemented with 1% NEAA and 1 mM sodium pyruvate. For all cell lines, the media were also complemented with 1% L-glutamine (Thermo Fisher Scientific, Naarden, The Netherlands), 10% heat-inactivated fetal bovine serum (FBS; Thermo Fisher Scientific, Naarden, The Netherlands), and 1% Antibiotic-Antimycotic (100 units/mL of penicillin, 100 µg/mL of streptomycin and 0.25 µg/mL of Fungizone; Thermo Fisher Scientific, Naarden, The Netherlands). Cells were incubated at 37 °C in a humidified atmosphere of 5% CO_2_.

### 2.2. Stable Transduction

The GIPZ™ lentiviral system (pLenti-GIII-CMV-GFP-2A-Puro) was chosen to overexpress *WWOX*, with Puro-Blank Lentivirus as control (Applied Biological Materials Inc., Richmond, BC, Canada). The cells were transduced in starvation medium with 8 µg/mL polybrene as the vehicle (Merck Life Sciences Sigma Aldrich, Darmstadt, Germany) and lentiviral particles at multiplicity of infection (MOI) = 1.5. After 24 h of incubation, the viral medium was changed to a complete medium. Antibiotic-based clone selection was performed using 1 µg/mL puromycin (Merck Life Sciences Sigma Aldrich, Darmstadt, Germany) after 72 h. The efficiency was confirmed by Western blot analysis.

### 2.3. Protein Extraction and Western Blot

Cells were lysed on ice with RIPA protein extraction buffer containing protease inhibitor cocktail (Santa Cruz Biotechnology Inc., Dallas, TX, USA). Concentration was determined using the Bradford method (Bio-Rad Laboratories, Hercules, CA, USA), and then 60 μg of proteins were resolved on 10% SDS-PAGE and transferred on polyvinylidene fluoride (PVDF) membranes (Merck Life Sciences Sigma Aldrich, Darmstadt, Germany). Staining with Ponceau red (Merck Life Sciences Sigma Aldrich, Darmstadt, Germany) confirmed proper transfer and was followed by membrane blocking using 5% non-fat milk in 1X TBST buffer (Merck Life Sciences Sigma Aldrich, Darmstadt, Germany) for 1 h at room temperature (RT). Membranes were then incubated for 18 h at 4 °C with a primary anti-WWOX antibody (PA5-29701, Thermo Fisher Scientific, Naarden, The Netherlands) diluted 1:1000 with 1% non-fat milk in 1X TBST solution. Subsequently, the membranes were washed with 1X TBST and incubated with goat anti-rabbit secondary antibody conjugated with alkaline phosphatase (Sigma-Aldrich) for 1 h at RT. Next, the membranes were washed with 1X TBST and developed with Novex^®^ AP chromogenic substrate (Invitrogen). Glyceraldehyde 3-phosphate dehydrogenase (GAPDH) was used as a reference protein (sc-59540, Santa Cruz Biotechnology Inc., Dallas, TX, USA). The relative protein amount was assessed with ImageJ (National Institutes of Health, Rockville, MD, USA) based on the integrated density of the bands. The assay was performed in duplicate (T98G, DBTRG-05MG) or triplicate (U251MG, U87MG).

### 2.4. Assessment of Redox Potential, Proliferation and Apoptosis

Cell viability, apoptosis, and proliferation assays were performed on a single 96-well plate. On the first day, cells were seeded in a complete medium at a concentration of 1.5 × 10^4^ cells per well and incubated for 24 h at 37 °C in 5% CO_2_. The complete medium was then changed into 100 µL starvation medium containing 10 µL of 5-bromo-2′-deoxyuridine (BrdU), and the cells were incubated for 24 h at 37 °C in 5% CO_2_. Subsequently, 10 µL of PrestoBlue reagent (Thermo Fisher Scientific, Naarden, The Netherlands) was added to the wells, and fluorescence (excitation at 550 nm and emission at 590 nm) was measured eight times at 10 min intervals (control wells: without cells and reagent). Apoptosis and proliferation were detected according to the manufacturer’s protocols using TUNEL assay (DELFIA^®^ DNA Fragmentation Assay; PerkinElmer, Vaughan, ON, Canada) and BrdU incorporation (DELFIA^®^ Cell Proliferation Kit; PerkinElmer, Vaughan, ON, Canada), respectively. The VICTOR X4™ Multilabel Plate Reader (PerkinElmer, Vaughan, ON, Canada) was used to measure fluorescence. The assay was performed in triplicate for each cellular variant.

### 2.5. Adhesion Assay

To investigate cell adhesion to collagen I and IV, laminin I and fibronectin with bovine serum albumin (BSA)-coated plate as a control, Corning^®^ BioCoat™ plates were used (BD Biosciences, San Jose, CA, USA). Cellular variants were seeded (1.5 × 10^5^ cells per well) in serum-free media and incubated for 4 h at 37 °C in 5% CO_2_. The cells were then washed three times with 1X PBS (Merck Life Sciences Sigma Aldrich, Darmstadt, Germany) and stained for 10 min using 0.1% crystal violet (Merck Life Sciences Sigma Aldrich, Darmstadt, Germany). Excess crystal violet was removed using distilled water. Extraction using 10% acetic acid was followed by the absorbance measurement at 560 nm with an Infinite F50 Tecan plate reader (Life Sciences, Mannedorf, Switzerland). The assay was performed in triplicate for each cellular variant.

### 2.6. Integrin-Mediated Cell Adhesion

The CHEMICON^®^ Alpha and Beta Integrin-Mediated Cell Adhesion Array kits were used to quantify cell surface subunit (α1, α2, α3, α4, α5, αV, β1, β2, β3, β4, β6) or heterodimer integrins (α5β1, αVβ3, αVβ5). After reaching full confluence, cells were harvested and added (1 × 10^5^) to the integrin antibody-coated and control wells. Plates were then incubated for 2 h at 37 °C. Unbound cells were washed away using 1X PBS (Merck Life Sciences Sigma Aldrich, Darmstadt, Germany) and stained for 10 min using crystal violet (Merck Life Sciences Sigma Aldrich, Darmstadt, Germany). Excess crystal violet was removed using distilled water. Extraction was followed by the optical density measurement at 540 nm with an Infinite F50 Tecan plate reader (Life Sciences, Mannedorf, Switzerland). The assay was performed in triplicate for each cellular variant.

### 2.7. Clonogenic Assay

Cells were seeded (1.5 × 10^3^/well) onto a 6-well plate in complete medium and cultured for 14 days (37 °C, 5% CO_2_), exchanging the media every three days. All cellular variants were then washed twice with 1X PBS and fixed with 4% paraformaldehyde (Merck Life Sciences Sigma Aldrich, Darmstadt, Germany) in PBS solution and stained with 0.005% crystal violet (Merck Life Sciences Sigma Aldrich, Darmstadt, Germany) for 15 min at RT. Excess crystal violet was removed, and colonies were counted using ImageJ software (National Institutes of Health, Rockville, MD, USA). The assay was performed in triplicate for each cellular variant.

### 2.8. Suspension Growth Test

Sterilized and heated 2.4% agar dissolved in water was used to evaluate the growth in the suspension. Two layers were poured onto 6-well plates. The bottom layer was a mixture of 750 µL of 2.4% agar and 1250 µL of medium (per well), then the plates were allowed to set the mixture. The upper layer consisted of 250 µL of 2.4% agar and 1750 µL of medium with 1 × 10^4^ cells per well. Cells were grown for 14 days (37 °C, 5% CO_2_) and fed by carefully adding 500 µL of medium dropwise every three days. Subsequently, the cell cultures in suspension were stained by adding 0.005% crystal violet (Merck Life Sciences Sigma Aldrich, Darmstadt, Germany) and incubated for 15 min at RT. The stained cells were counted using ImageJ (National Institutes of Health, Rockville, MD, USA). This assay was performed in triplicate for each cellular variant.

### 2.9. Three-Dimensional Culture Growth Assay

This assay used the Geltrex matrix (Thermo Fisher Scientific, Naarden, The Netherlands), comprising a basement membrane matrix including laminin, collagen IV, entactin/nidogen, and heparin sulfate proteoglycan, which are involved in the tissue organization. Geltrex was thawed on ice, and the solidified 2 mm layer was added to a 96-well plate. The cells were then plated in a complete medium on the surface of the three-dimensional protein matrix at a density of 1.5 × 10^3^ cells/well and were then incubated for 15 days (37 °C, 5% CO_2_). Subsequently, the cells were observed under the inverted light microscope. This assay was performed in triplicate for each cellular variant.

### 2.10. Invasion Assay

Corning^®^ BioCoat™ Matrigel^®^ Invasion Chambers with 8 μm polyester membranes (BD Biosciences, San Jose, CA, USA) were used to assess the invasiveness potential [58,59]. The cells were suspended in serum-free medium, seeded (2 × 10^5^/well) onto the inner compartment of inserts, and the complete medium was added to each lower chamber. Following the incubation for 48 h at 37 °C, 5% CO_2_, the cells on the upper surface of the microporous membrane were removed with cotton swabs. The cells on the lower surface of the membrane were subsequently stained with 0.1% crystal violet (Merck Life Sciences Sigma Aldrich, Darmstadt, Germany). For the extraction process, 200 μL of 10% acetic acid (Merck Life Sciences Sigma Aldrich, Darmstadt, Germany) per well was used. The absorbance of extracts was measured using an Infinite F50 Tecan plate reader (Life Sciences, Mannedorf, Switzerland). The assay was performed in triplicate for each cellular variant.

### 2.11. Cumulative Degradation of Gelatin Layer

The QCM™ Gelatin Invadopodia Assay kit (Millipore, Burlington, MA, USA) was used to evaluate gelatin layer degradation. Glass coverslips in a standard 24-well plate were treated with 250 µL poly L-lysine (Millipore’s Part No. CS207800), rinsed three times with sterile Dulbecco’s PBS (DPBS; Merck Life Sciences Sigma Aldrich, Darmstadt, Germany), cross-linked with 250 µL glutaraldehyde (Millipore’s Part No. CS207801) and rinsed again with DPBS. Then, the coverslips were coated with 200 µL of fluorescent gelatin (1:5 dilution of Cy3-Gelatin (Millipore’s Part No. CS207803) with unlabeled gelatin (Millipore’s Part No. CS207805)) and incubated at RT for 10 min, protected from light. The plates with coverslips were then rinsed with DPBS and incubated with 70% ethanol for 30 min without light. Plates were rinsed three times with DPBS, and DMEM media was added to the well at RT for 30 min before cell plating. The cells were seeded at a density of 5 × 10^4^ in fresh growth media at 500 µL/well. After incubating the cells at 37 °C for a further 24 h, they were fixed in 3.7% paraformaldehyde (PFA) using 250 µL/well, incubated at RT for 30 min, protected from light and then rinsed twice with DPBS. Before staining, the plates were rinsed twice with fluorescent staining buffer (DPBS with 2% BSA + 0.25% Triton X-100). Next, 100 µg/mL FITC-phalloidin (Millipore’s Part No. CS207821) and 100 µg/mL DAPI (Millipore’s Part No. 90229) were diluted in fluorescent staining buffer to a final concentration of 2 µg/mL and 1 µg/mL, respectively. The cells were then incubated in 200 µL of staining solution for 1 h at RT and protected from light. After incubation, the staining solution was removed, and cells were rinsed twice with DPBS. Coverslips were inverted onto glass slides for imaging. The slides were observed under the 20 X objective of the IX3-FP confocal microscope (Olympus, Tokyo, Japan). The degraded gelatin area was quantified using ImageJ software (National Institutes of Health, Rockville, MD, USA). Experiments were completed in triplicate for each cellular variant.

### 2.12. Gelatin Zymography Assay

Cells were seeded on 6-well plates at a density of 2 × 10^5^ cells/well and cultured in a complete medium to obtain 80% confluence. Subsequently, the complete medium was exchanged for a serum-free medium. After 48 h of incubation (37 °C, 5% CO_2_), the starvation medium with detached cells was collected and centrifuged at 400× *g* for 5 min at 4 °C. The Qubit Protein Assay was used to measure the protein concentration on a Qubit 2.0 Fluorometer (Thermo Fisher Scientific, Naarden, The Netherlands); subsequently, 3 µg of proteins and ladder were loaded onto the 10% polyacrylamide co-polymerized with 0.1% *w*/*v* gelatin (Merck Life Sciences Sigma Aldrich, Darmstadt, Germany). The gels were washed (3 × 10 min) with 2.5% Triton X-100 (Merck Life Sciences Sigma Aldrich, Darmstadt, Germany) and incubated overnight in a developing buffer (0.5 M Tris-HCl, 2 M NaCl, 50 mM CaCl2, pH 7.5) at 37 °C. Subsequently, staining of the gels was performed using Coomassie Brilliant Blue R-250 (Merck Life Sciences Sigma Aldrich, Darmstadt, Germany), and the zymogram was then washed with destaining solution (methanol: acetic acid: water, 3: 1: 6) to obtain clear bands at the areas where gelatin was degraded. The activity of matrix metalloproteinases (MMPs) was determined using ImageJ software (National Institutes of Health, Rockville, MD, USA). The assay was performed in triplicate for each cellular variant.

### 2.13. Bioinformatics Analysis

Data on gene expression, mutations, structural variants, and copy number alterations (CNAs) were acquired from the Cancer Cell Line Encyclopedia (CCLE) of the Broad Institute (2019 version) available at cBioPortal [60,61]. The user-defined list of cell lines (T98G, DBTRG-05MG, U251MG, and U87MG) was selected with mRNA expression Z-score threshold set at ± 0.6. The list of genes used as a cBioPortal query was created to reveal a possible cause of the DBTRG-05MG discrepancies found in the small subset of in vitro assays. The list of mutated genes in DBTRG-05MG (acquired from COSMIC Cell Line Gene Mutation Profiles [62,63]) was initially filtered to exclude mutated genes that are also present in T98G, U251MG, and U87MG, and then using the list of known WWOX-related genes from our previous glioblastoma research [64]. This approach revealed 56 mutated genes (of 634) that might be of interest both in terms of DBTRG-05MG exclusiveness and relation to WWOX. A manual investigation of these genes indicated an interesting observation for *CPNE2* and *USP25*; their expression, mutation, and CNA data were visualized in cBioPortal. *CPNE2*, *USP25*, and their superior transcription factor TARDBP were correlated with WWOX using the TIMER2.0 database (employed via the “Gene_Corr” module among GBM samples and with purity adjustment) [65]. The ratio between the expression of *WWOX* and *TARDBP* for all cell lines was calculated using the reads per kilobase million (RPKM) values acquired from the “Download” tab, which is the last in the cBioPortal query. TARDBP binding sites within the *CPNE2* and *USP25* genes were visualized using the Gene Transcription Regulation Database (GTRD) [66]. Pathways that downregulate or upregulate TARDBP were determined using the Signaling Pathway Enrichment using Expression Data set version 2 (SPEED2) [67]. A hypothetical explanation that could clarify the cause of discrepancies found in DBTRG-05MG was visualized with the help of BoxPlotR [68] and Inkscape (https://www.nihlibrary.nih.gov/resources/tools/inkscape).

### 2.14. Statistical Analysis

Statistica v13.1 (StatSoft, Tulsa, OK, USA) and GraphPad Prism v8 (GraphPad Software, San Diego, CA, USA) were employed for statistical analysis. Levene’s test was used to test the homogeneity of variance. The Shapiro–Wilk test allowed us to determine the normality of distribution. An unpaired t-test or Wilcoxon test evaluated statistical relevance. Results with a *p*-value less than 0.05 were considered statistically significant.

## 3. Results

### 3.1. Confirmation of the Obtaining of Stable Transductants

The efficiency of the performance of the stable transduction was evaluated by the Western blotting technique. The relative amount of protein indicated the overexpression of *WWOX* in T98G (Figure 1A), DBTRG-05MG (Figure 1B), U251MG (Figure 1C), and U87MG (Figure 1D) glioblastoma cell lines in comparison with “CONTROL” variants. The mean relative protein level of “WWOX” was significantly higher (20.93 ± 0.8359 for T98G; 11.72 ± 0.8161 for DBTRG-05MG; 0.86 ± 0.3365 for U251MG; and 12.50 ± 1.6540 for U87MG), than for “CONTROL” (7.97 ± 0.9851, *p* < 0.0001; 0.80 ± 0.0379, *p* < 0.0001; 0.07 ± 0.0346, *p* = 0.0155; and 1.23 ± 0.2234, *p* = 0.0003; respectively). Accurate densitometric values are also shown in Figure 1E.

### 3.2. WWOX Intensified the Apoptosis but Reduced Mitochondrial Redox Potential

The Triplex assay allows for the evaluation of the apoptotic, proliferative, and mitochondrial redox potential of the tested cells (Figure 2). Overexpression of *WWOX* significantly intensified the programmed cell death of T98G (*p* = 0.042), DBTRG-05MG (*p* = 0.009), and U87MG (*p* = 0.021) cells (Figure 2A). In terms of proliferation, the “WWOX” variant significantly reduced this biological process in T98G (*p* = 0.0041) and U87MG (*p* = 0.0264) cells, whereas it increased in DBTRG-05MG (*p* < 0.0001) (Figure 1B). In each analyzed cell line, *WWOX*-overexpressing cells significantly diminished the mitochondrial redox potential, except for the 10-min measurements for the T98G and DBTRG-05MG cells. The most statistically significant decrease in cell viability was observed for the U87MG cell line compared with the others. Statistical significance for individual comparisons and measurement time points in redox assay are visualized in Figure 2C.

### 3.3. Overexpression of WWOX Decreased the Adhesion to Collagen I, Collagen IV, Fibronectin, and Laminin I

Adhesion to the extracellular matrix (ECM) proteins evaluated in this study, i.e., collagen I, collagen IV, fibronectin, and laminin I, was congruent in the tested cell lines (Figure 3A). In the U251MG cell line, the variant with increased *WWOX* expression significantly reduced adhesion to all tested ECM proteins (*p* = 0.009 for collagen I; *p* < 0.0001 for collagen IV; *p* = 0.049 for fibronectin; and *p* = 0.035 for laminin I). The “WWOX” variant of the U87MG cell line also showed significantly decreased adhesion to the collagen I (*p* = 0.024), collagen IV (*p* < 0.0001), fibronectin (*p* = 0.047), and laminin I (*p* = 0.006). BSA served as a negative control in this experiment.

Integrins mediate cell attachment to ECM proteins and the results obtained in the above adhesion test were consistent with integrin panels. In the alpha integrins panel (Figure 3B), the “WWOX” variant significantly decreased the following subunits or heterodimers: α1 (*p* = 0.002 for T98G; *p* = 0.006 for DBTRG-05MG), α2 (*p* = 0.042 for T98G; *p* = 0.045 for DBTRG-05MG; *p* = 0.012 for U251MG; *p* = 0.015 for U87MG), α3 (*p* = 0.022 for U87MG), α4 (*p* = 0.029 for U251MG), α5 (*p* = 0.032 for T98G; *p* = 0.044 for DBTRG-05MG; *p* = 0.016 for U251MG; *p* = 0.033 for U87MG), αV (*p* = 0.0002 for T98G; *p* = 0.022 for DBTRG-05MG; *p* = 0.036 for U251MG; *p* = 0.004 for U87MG), and αVβ3 (*p* = 0.014 for U251MG; *p* = 0.0196 for U87MG).

Additionally, considering the beta integrins panel, the variant with *WWOX* overexpression diminished β1 (*p* = 0.028 for T98G; *p* = 0.009 for U251MG; *p* = 0.008 for U87MG), β4 (*p* = 0.006 for T98G), β6 (*p* = 0.039 for T98G; *p* = 0.032 for DBTRG-05MG), αVβ5 (*p* = 0.044 for T98G; *p* = 0.017 for U251MG; *p* = 0.0007 for U87MG), and α5β1 (*p* = 0.027 for T98G; *p* = 0.0004 for DBTRG-05MG; *p* = 0.035 for U251MG; *p* = 0.015 for U87MG). These data are visualized in Figure 3C.

### 3.4. WWOX Acts in an Opposite Manner in DBTRG-05MG: Enhancing Colony Forming Abilities, Suspension Growth, and Increasing the Size of Dimensional Spheres

Colony formation assay verified the self-renewal capacities of T98G, DBTRG-05MG, U251MG, and U87MG glioblastoma cell lines (Figure 4A). *WWOX*-overexpressing cells significantly decreased the number of colonies in T98G (*p* < 0.0001), U251MG (*p* = 0.0245), and U87MG (*p* = 0.0275), whereas these were increased in DBTRG-05MG (*p* = 0.0216) glioblastoma cells.

These results are in line with observations from growth in soft agar (Figure 4B); the statistically significant effects were obtained for T98G (*p* = 0.0004) and DBTRG-05MG (*p* = 0.0337). However, both U251MG and U87MG cells were unable to grow in suspension culture.

The disparities for DBTRG-05MG were also noticed in the ECM three-dimensional growth (Figure 4C). Considering the differences between “WWOX” and “CONTROL” variants in individual cell lines, the effect was more visible in the quantity of colonies for T98G and U87MG than in the size of the spheres for U251MG and DBTRG-05MG. For the latter cell line, the *WWOX* overexpression showed the opposite outcome to the other cell lines, increasing the size of the cellular clusters.

### 3.5. WWOX Acted Consistently as a Suppressor of Invasion, Gelatin Degradation, and MMP-9

Results of the invasiveness assay in all examined cell lines are statistically significant (Figure 5A) for T98G, DBTRG-05MG, U251MG, and U87MG (*p* = 0.0149, *p* < 0.0001, *p* < 0.0001, *p* < 0.0001, respectively). Observations of the *WWOX* overexpression effect were consistent in all cases, indicating a reduction of invasiveness in the tested cell lines.

The second method for analyzing cellular invasion was the application of a gelatin layer and quantifying the relative number of cells that have traversed the layer (Figure 5B). The results of this assay are corresponding and indicate that *WWOX*-overexpressing cells significantly diminished the invasion in DBTRG-05MG (*p* = 0.0392), U251MG (*p* = 0.0430), and U87MG (*p* = 0.0184) cell lines.

The outcome of the gelatin zymography assay (Figure 5C) revealed that the MMP-9 activity for all comparisons was not only reduced in “WWOX” variants but also coherent with the two above assays. The statistical significances for the individual cell lines were as follows: *p* = 0.0478 for T98G, *p* = 0.0022 for DBTRG-05MG, *p* = 0.0009 for U251MG, and *p* = 0.0339 for U87MG.

## 4. Discussion

The location of *WWOX* in a chromosomal region that is frequently altered in various cancers has prompted research to determine its role in carcinogenesis. Moreover, literature data on in vivo studies suggest that *WWOX* in murine models has a crucial role in biological processes such as growth or metabolism, and that its loss leads to enhanced cancer stemness [69,70]. Recent molecular and clinical analyses of WWOX functions have emphasized its role in the modulation of signaling pathways related to cancer promotion, metabolism, CNS development, and neurological diseases [10,70,71,72]. Furthermore, our previous in silico study proved that the WWOX is associated with complex protein networks, highlighting its direct and indirect function in maintaining cell homeostasis in GBM [64]. Because the majority of the currently available in vitro research does not include multiple cellular GBM models or a wide range of investigated biological assays, our research aimed to identify the main processes by which *WWOX* exhibits anticancer properties in GBM, considering the phenotypic heterogeneity of T98G, DBTRG-05MG, U251MG, U87MG cell lines.

The *WWOX*-induced changes in apoptosis, viability, and proliferation are generally coherent except for the DBTRG-05MG cell line. In brief, WWOX intensified cellular apoptosis but decreased mitochondrial redox potential. As for proliferation, it was found to be reduced in U87MG and T98G cells but elevated in DBTRG-05MG. Discrepancies for DBTRG-05MG might be attributed to the specific features of these cells (among four investigated cell lines), e.g., the presence of a BRAF V600E mutation [73,74]. It is known that this alteration can lead to the maintenance of proliferation, transformation, and tumorigenicity [75] and that multiple proliferation-related signaling pathways are simultaneously active when V600E is present [76]. One can speculate that not all V600E-dependent proliferation cues are diminished following *WWOX* overexpression in DBTRG-05MG. The topic of BRAF alterations is also relevant in the GBM context. For instance, the study by Andrews et al. provided the rationale for adjuvant clinical trials of BRAF inhibitor treatment in V600-mutant glioma [77]. Nevertheless, Kośla et al. observed that the T98G cells in the *WWOX*-overexpressing variant were less proliferative, certifying the results of our present study [33]. With regard to the elevated apoptosis, it is in line with a widespread opinion that WWOX functions as a pro-apoptotic protein [37], inter alia, owing to the pro-apoptotic D3 tail in its structure [78]. Further evidence can be found in the direct physical interaction with p53 and p73 via the first tryptophan domain, as well as the impact on the tumor necrosis factor (TNF) apoptotic pathway [79]. As for cell viability, our previous research on bladder cancer also demonstrated that WWOX decreases mitochondrial redox potential, with the explanation for this possibly to be found in the link between TNF and reactive oxygen species [29].

Adhesiveness to ECM proteins revealed very promising observations for two cell lines. In brief, *WWOX* overexpression decreased adhesion to collagen I and IV, fibronectin, and laminin I in U251MG and U87MG cell lines. The decrease in adhesion to collagens is most likely due to changes in the α2 and β1 integrin levels; the α2β1 integrin is a default collagen-binding integrin [80]. Furthermore, the similarity between the two aforementioned cell lines is reflected in data for the fibronectin that is dependent on α5β1 integrin [81], the differences for which are coherent in α5 and β1, as well as combined α5β1. However, the same change in adhesion to laminin (following *WWOX* overexpression) might be due to the various α integrins in U251MG and U87MG. This seems to be dependent on α4β1 for the former, whereas, for the latter cell line, it is related to α3β1. These integrins are known receptors for laminin [82,83]. The results of the present research also complement the data from the study by Kośla et al., wherein the *WWOX* overexpression decreased the adhesion of T98G cells to collagen I, collagen IV, and fibronectin [33], which is now certified respectively in the change of α2β1, α1β1, and α5β1. Other literature data indicate that the level of α2 and α3 integrins were significantly decreased in the *WWOX*-transfected cells [84]. A reduced level of α2 integrin is present in “WWOX” variants of all cell lines that we investigated, whereas the decreased α3 is visible only in U87MG. The expression of integrins is upregulated in various cancers, including GBM, for which αVβ3 and αVβ5 were the first to be identified in gliomas as differentially expressed in comparison with normal brain tissues [85]. Malric et al. have stated that overexpression of integrins α3, α5, αV, β1, and β3, is associated with poor prognosis in glioblastoma patients [86]. The decrease in the level of three out of these five integrins (α3, α5, β1) allowed us to explain the changes in adhesion to various ECM proteins during *WWOX* overexpression. Given the role of integrins in potentiating GBM invasion [87], some WWOX-related changes in the invasiveness are expected. These will be discussed in an antepenultimate paragraph.

Tumor growth of DBTRG-05MG cells was aggravated following *WWOX* overexpression, but the opposite was found in T98G, U251MG, and U87MG. In the last three cell lines, *WWOX* overexpression decreased the number of colonies, as well as the anchorage-independent and three-dimensional growth. Both the quantity of colonies, as well as the three-dimensional growth were previously investigated by us in bladder cancer using both *WWOX*-overexpressing and *WWOX*-downregulating variants [88], which indicated that WWOX reduces clonogenicity and affects the formation of cellular aggregates. This certifies our present results in the GBM context. Regarding the anchorage-independent growth, our observations can be confirmed by another study that investigated *WWOX* overexpression and knockdown in cervical cancer [89]. Nevertheless, the reason that DBTRG-05MG presented different clonogenicity and anchorage-independent growth is enigmatic. Searching through possible explanations, we noticed that, besides the difference in BRAF V600 mutational status (V600E is present in DBTRG-05MG but not in T98G, U251MG, and U87MG), the DBTRG-05MG cell line generally contains more mutations, as indicated by the COSMIC cell line gene mutation profiles [62,63]. Compared with U251MG and U87MG, the number of mutations in DBTRG-05MG is doubled, whereas relative to T98G, the DBTRG-05MG possesses 1.6 times more mutations. However, we cannot identify the difference in mutational status in genes known for their biological relevance in GBM tumorigenesis, such as the *IDH1* gene that is unaltered in all investigated cell lines according to the CCLE. Nevertheless, based on the same database, DBTRG-05MG is the only cell line that harbors a mutation in another member of the isocitrate dehydrogenase family—*IDH3G*. More attention should be given to DBTRG-05MG in future research.

It is currently challenging to mechanistically elucidate how *WWOX* decreases the invasiveness of GBM without referring to other tumors. Interesting, invasion-related findings were found both upstream and downstream of WWOX. First, miR-670-5p was found to decrease WWOX expression, promoting migration or invasion [90]. Another example of upstream WWOX regulation is activated Cdc42-associated kinase 1 (ACK1) signaling that acts negatively on WWOX expression but positively on AKT signaling. Knockdown of ACK1 in hepatocellular carcinoma repressed, e.g., invasion and migration and downregulated MMP-9. Similar research is needed in the glioblastoma context since the invasion and MMP-9 expression were also reduced in our present study following *WWOX* overexpression. At this point, ACK1 might be a missing puzzle, especially because ACK1-AKT signaling was found to promote glioma tumorigenesis [91]. Considering downstream regulation, WWOX inhibited the expression of RUNX2 and its target gene *MMP9*, reducing cancer invasiveness [92]. This might be due to the known direct interaction between WWOX and SMAD3, which reduces the occupancy of the latter on target gene promoters [93]; RUNX2 is one such gene [94]. Alternatively, WWOX can directly bind to RUNX2, modulating its transcriptional activity [95]. Another study by Xu et al. proposed the anti-invasive potential of WWOX via enhancement of ELF5 activity, thus decreasing *SNAIL1* and further upregulating *CDH1* [96]. Finally, an interesting study about lncRNA WWOX-AS1 seems to certify the above data. Qu et al. indicated the positive correlation between WWOX and WWOX-AS1 [97], suggesting that the latter is one of the activator antisense lncRNAs that regulate their sense transcript in a positive manner [98,99]. Because WWOX-AS1 was found to inhibit proliferation, migration, and invasion, one can speculate that the impact of positively correlated WWOX was similar or at least did not act oppositely. Although the authors clearly stated that they did not determine the exact molecular mechanism, they hypothesized that it might be associated with RUNX2 or genes activated by this transcription factor [97].

Collectively, our study demonstrates that WWOX intensifies apoptosis, suppresses proliferation and viability, diminishes adhesion to various ECM proteins, reduces tumor growth and the quantity of colonies, and reduces the invasiveness of GBM. These findings apply to T98G, U251MG, and U87MG cell lines, whereas particular attention should be given to DBTRG-05MG cells that presented discrepancies in tumor proliferation and growth. Although this study cannot entirely capture the reason for divergency, the fact of possessing much more mutations (including those that are biologically relevant, e.g., BRAF V600E) in DBTRG-05MG might be implicated in this issue. All genes from previous paragraphs, which were identified when discussing our results with the use of literature, are also summarized in tabular form to collate the data on their mutational status, CNAs, and mRNA expression (Appendix A). Additionally, one could search for a more WWOX-related explanation to conform with cellular variants developed in this study. To provide such, we performed the preliminary bioinformatics analysis to disclose any DBTRG-05MG-specific mutations that could contribute to opposite findings in this cell line. Following the identification of WWOX-related genes that are only mutated in DBTRG-05MG, the manual investigation revealed an interesting observation for *CPNE2* and *USP25* (Figure 6A). Not only was the expression of these genes found to be the lowest (*CPNE2*) or the highest (*USP25*) in DBTRG-05MG relative to other investigated cell lines (Figure 6B), but also the in-frame mutation of *USP25* (V809dup) was present exclusively in DBTRG-05MG and was one of the cell line-specific mutations that could be related to CNAs. Although V809dup is currently of unknown significance according to cBioPortal, it is theoretically possible that in-frame mutation leads to overexpression or activation [100,101], and may result in an abnormal protein product [102]. Even if not in this case, the native 2.5-fold higher expression of *USP25* in DBTRG-05MG is evident when compared with T98G, U251MG, and U87MG. Furthermore, the implication of *USP25* and *CPNE2* in a single molecular process prompted us to proceed with this explanation. Namely, it appears that these genes regulate cell surface abundance of epidermal growth factor receptor (EGFR): *USP25* was found to decrease EGFR degradation and internalization [103], whereas *CPNE2* was related to the decreased EGFR surface abundance [104]. Downregulating the cell surface EGFR is a major process in the attenuation of EGFR signaling [105]. In GBM, the overexpression of *EGFR* is a striking feature that is observed in approximately 40% of tumors [106]. Since *CPNE2* and *USP25* are in the list of known WWOX-related genes from our previous glioblastoma research [64], but might not be the interacting WWOX protein partners (the lack of literature complicates the inference), we decided to use the WWOX interactome [36] to identify the WWOX-interacting transcription factor that could regulate expression of these genes. It turned out that TARDBP is one of these transcription factors; as it is positively correlated with WWOX (Figure 6C) it might be upregulated by, e.g., the Notch signaling (Figure 6D) that was previously found to be modulated by WWOX in glioblastoma [33]. Moreover, the binding sites of TARDBP are present in *CPNE2* and *USP25* genes that also positively correlate with *WWOX* (Figure 6E). It is worth mentioning that *TARDBP* has been linked to proliferation [107,108]. Furthermore, the ratio (Ri) of *WWOX* expression relative to *TARDBP* is the lowest in DBTRG-05MG (Ri = 0.0165) compared with the T98G (Ri = 0.0392), U251MG (Ri = 0.0731), and U87MG (Ri = 0.0391). Altogether, we hypothesized that, due to the higher level of *USP25* and the lower level of *CPNE2* in DBTRG-05MG following WWOX overexpression (marked with red in Figure 6F), the expression of *USP25* in this cell line may be the first to passes through the hypothetical threshold, rendering an imbalance between signals that regulates the EGFR surface abundance in favor of decreased internalization. In turn, this may lead to elevated proliferation; however, further investigation with the use of, e.g., high-throughput sequencing would allow the verification of these statements in the future. A short synopsis of our WWOX-related explanation is visualized in Figure 6F,G.

As with the majority of studies, our research is subject to limitations. For example, the in vitro findings often fail to extrapolate the cellular conditions in an organism, so future in vivo validation is recommended. Moreover, an attempt should be made to explain the discrepancies observed in DBTRG-05MG vs. the other cell lines with the use of high-throughput sequencing, which we plan to undertake. Lastly, in vitro cancer research is frequently supported by insights into the extracellular vesicles, such as the exosomes that are also highly heterogenous, similar to cancer cell lines [109,110]. To the best of the authors’ knowledge, research on WWOX in exosomes is scarce, emphasizing the need for relevant experiments performed by the scientific community at the earliest occasion. Nevertheless, promising results from a study by Xu et al. reveal that *WWOX* mRNA is present in exosomes isolated from the serum of osteosarcoma patients with good and poor chemotherapeutic response, as well as healthy controls [111]. It is worthy of note that the level of *WWOX* mRNA in exosomes from healthy controls was similar to that of good responders; at the same time, there was a statistically significant decrease of exosomal *WWOX* mRNA in poor responders compared with good responders. Exosomal mRNAs may regulate the stability, localization, and translational activity of mRNAs in target cells [112]. Thus, delivering various amounts of *WWOX* mRNA might entail diverse outcomes, which is a concept worth investigating in parallel to more classical mechanisms by which exosomes perform, e.g., delivering micro RNA (miRNA) or long non-coding RNA (lncRNA) [113]. Evidence of the ability of miRNAs and lncRNAs to regulate *WWOX* expression [114,115,116,117] definitely contributes to the study of RNA interactome complexity, rendering further research reasonable.

## 5. Conclusions

Our study indicates that *WWOX* intensifies apoptosis, suppresses proliferation and viability, diminishes adhesion to various ECM proteins, reduces tumor growth and the quantity of colonies, and reduces the invasiveness of GBM. These findings apply to the T98G, U251MG, and U87MG cell lines, whereas particular attention should be given to DBTRG-05MG cells that presented discrepancies in tumor proliferation and growth. Independently of the cause (i.e., if our scenario eventually does not turn out the be the most relevant), we presume that DBTRG-05MG cells are successfully opposed by increased apoptosis and viability, as well as by reduced invasion, the extensiveness of which affects the recurrences and short survival that entail the inferior outcomes of patients. To conclude, this research demonstrates that, even in various cell line-specific circumstances, WWOX exhibits its anti-GBM nature mainly via reductions in cell viability and in the invasiveness of glioblastoma.

## Figures and Tables

**Figure 1 biology-12-00465-f001:**
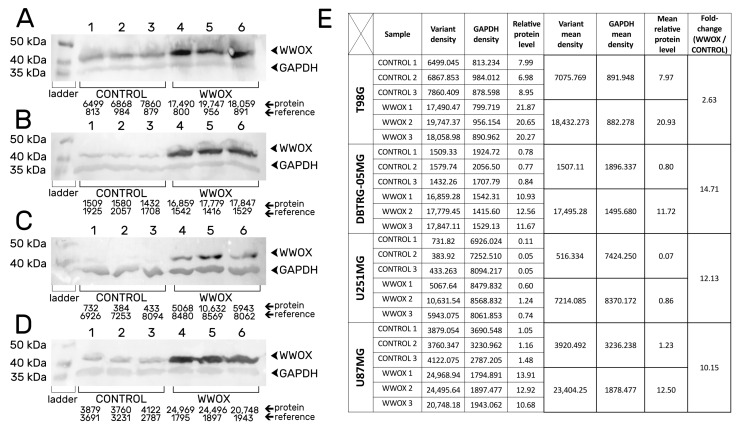
Stable transduction confirmation on the protein level for (**A**) T98G, (**B**) DBTRG-05MG, (**C**) U251MG, and (**D**) U87MG cell lines. (**E**) The tabular form of densitometric values.

**Figure 2 biology-12-00465-f002:**
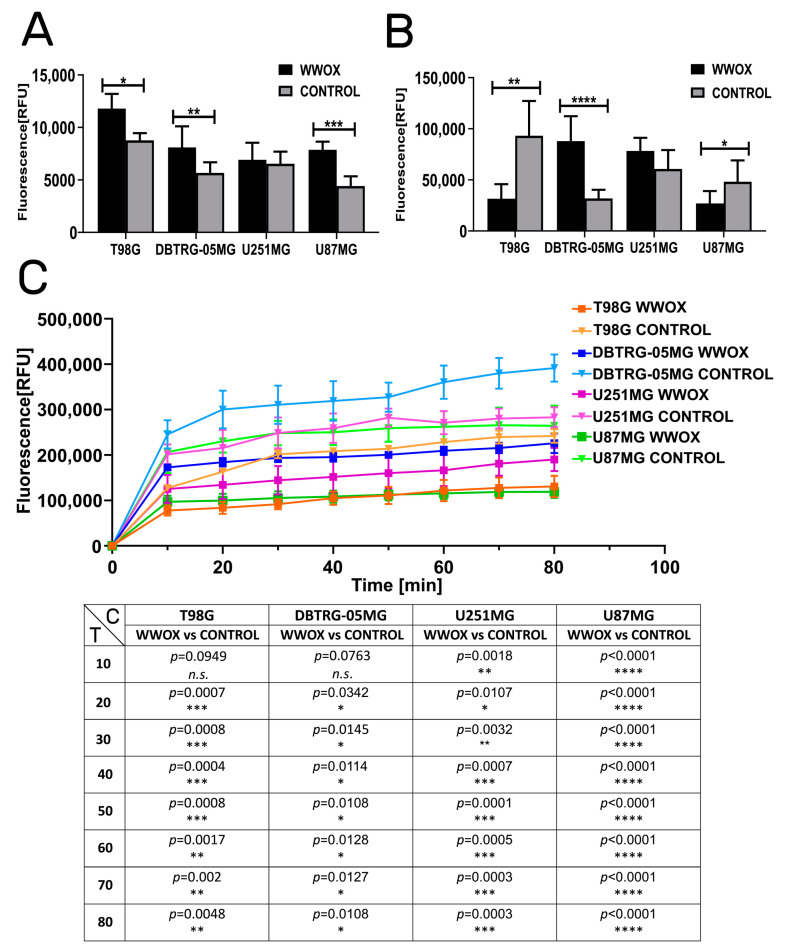
Determining the apoptotic, proliferative, and mitochondrial redox potentials. Plots collect data for T98G, DBTRG-05MG, U251MG, and U87MG cell lines. (**A**) Apoptosis. (**B**) Proliferation. (**C**) Mitochondrial redox potential. T—Time [min]. C—Comparison. *p* < 0.05 (*), *p* < 0.01 (**), *p* < 0.001 (***), *p* < 0.0001 (****).

**Figure 3 biology-12-00465-f003:**
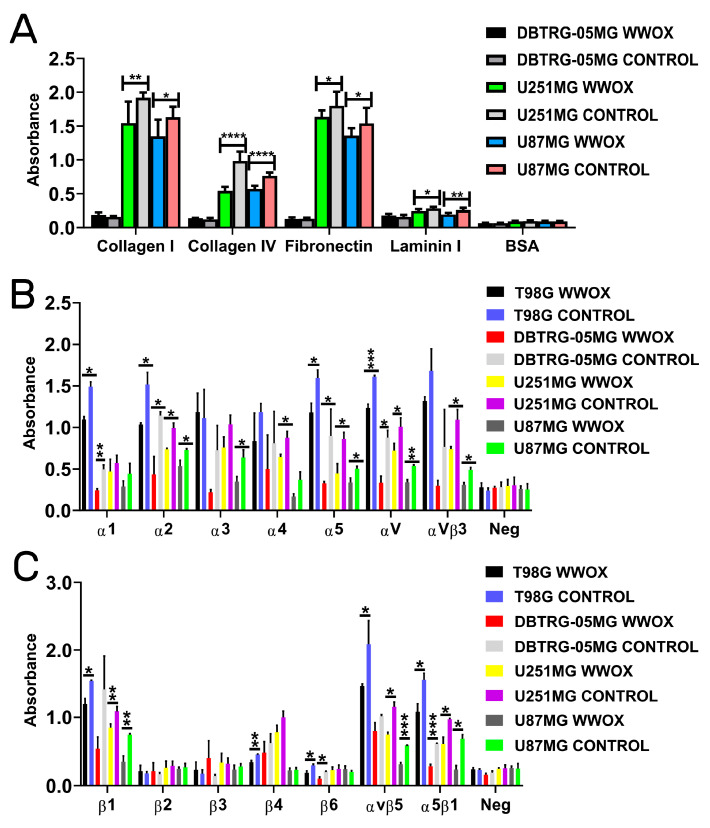
Adhesiveness to ECM proteins together with panels of α and β integrins. (**A**) Adhesion to ECM proteins. BSA—negative control. (**B**) α-subunits of integrins. Neg—negative control. (**C**) β-subunits of integrins. Neg—negative control. *p* < 0.05 (*), *p* < 0.01 (**), *p* < 0.001 (***), *p* < 0.0001 (****).

**Figure 4 biology-12-00465-f004:**
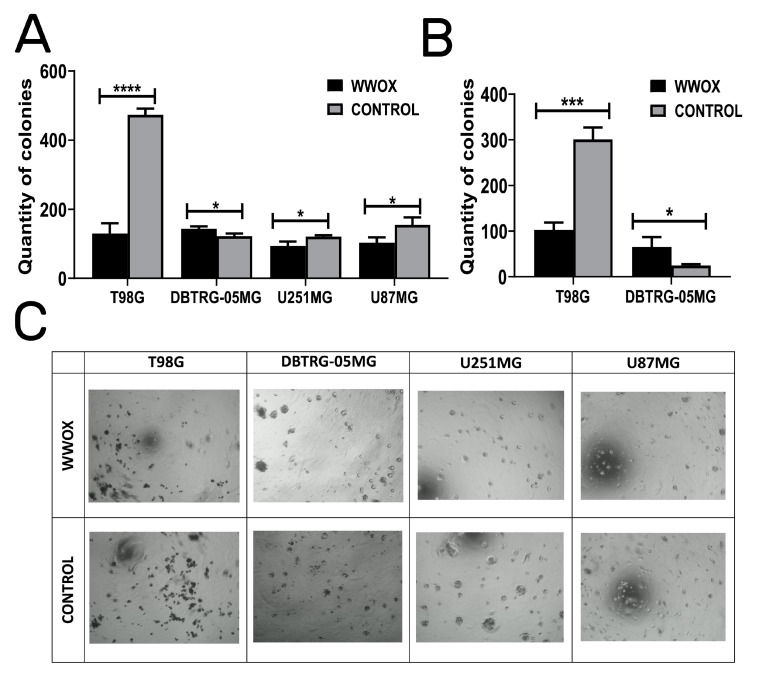
Observation of changes in the number of colonies and their anchorage-independent or three-dimensional growth. (**A**) Clonogenicity. (**B**) Anchorage-independent growth. (**C**) Three-dimensional culture in Geltrex. *p* < 0.05 (*), *p* < 0.001 (***), *p* < 0.0001 (****).

**Figure 5 biology-12-00465-f005:**
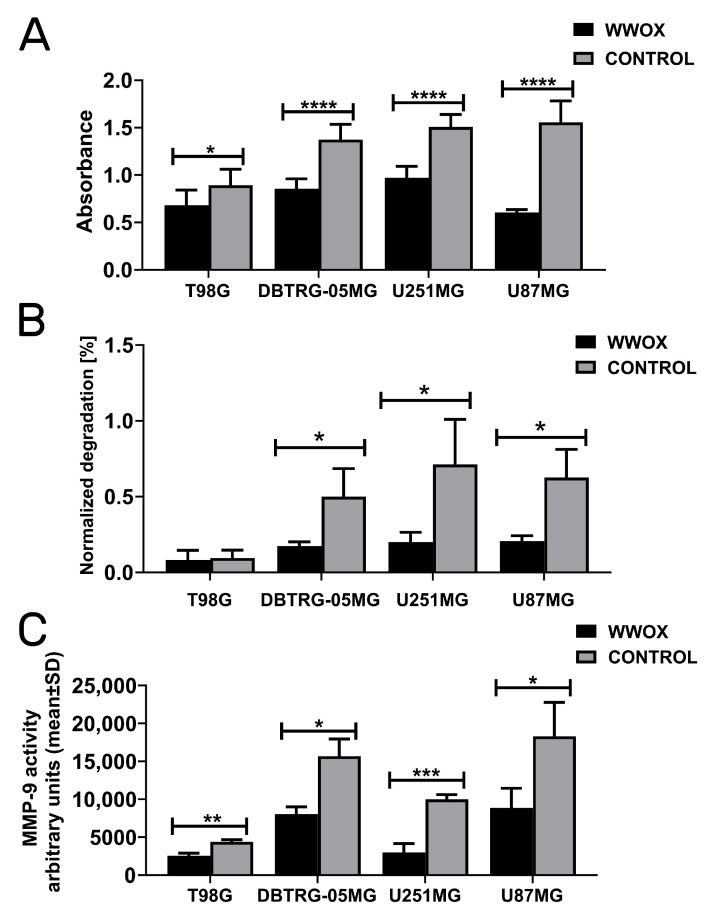
Determining the invasiveness, gelatin degradation, and MMP-9 activity. (**A**) Invasion assay. (**B**) Gelatin-layer degradation. (**C**) MMP-9 activity. *p* < 0.05 (*), *p* < 0.01 (**), *p* < 0.001 (***), *p* < 0.0001 (****).

**Figure 6 biology-12-00465-f006:**
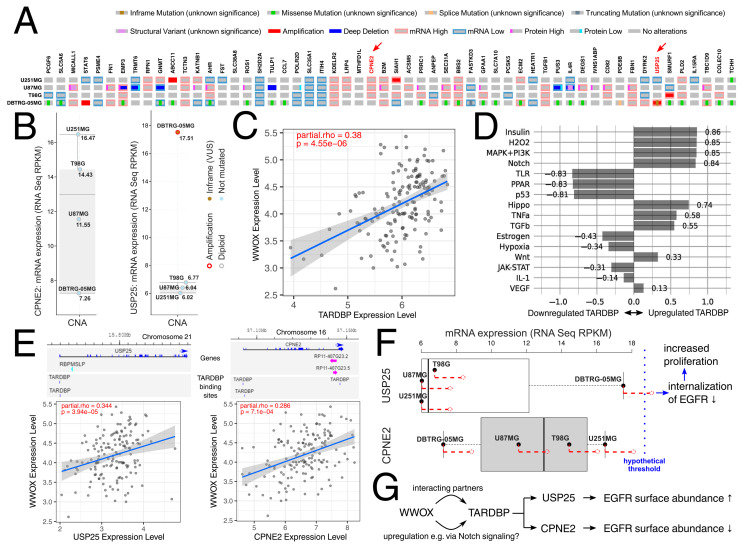
The possible explanation of discrepancies between DBTRG-05MG and other investigated cell lines. (**A**) Genomic profile of WWOX-related genes that are acquired from the full list of mutated genes in the DBTRG-05MG cell line. Interesting observations for *CPNE2* and *USP25* are marked with a red arrow. (**B**) Expression and copy number alterations of *CPNE2* and *USP25*. (**C**) Correlation between *WWOX* and *TARDBP*. (**D**) Signaling pathway enrichment for *TARDBP*. (**E**) Correlation between *WWOX* and *USP25* or *CPNE2*, with available TARDBP binding sites for the last two genes. (**F**) Differences in the level of *USP25* and *CPNE2*. Following *WWOX* overexpression (marked with red), the expression of *USP25* in the DBTRG-05MG cell line may be the first to pass through the hypothetical threshold, leading to decreased EGFR internalization and a subsequent increase in proliferation. (**G**) A short synopsis of the described explanation. CNA—Copy number alteration. VUS—Variant of unknown significance.

## Data Availability

The datasets supporting the conclusions of this article are included within the article (and its additional files).

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
