# Peer review of "Antineoplastic Nature of *WWOX* in Glioblastoma Is Mainly a Consequence of Reduced Cell Viability and Invasion"

_biology, 2023, doi:10.3390/biology12030465_

Round 1
Reviewer 1 Report
The submitted manuscript is of a very high quality, only minor corrections are needed. The detailed list of suggestions can be found below.
In the introduction it should be mentioned that WWOX is also known as human accelerated region 6.
Lines 43, 44-what is the first one?
Line 100, was the humidity measured?
Line 198, a reference is needed there.
Lines 252-256, what was the software used for ANOVA analysis?
Line 388, it should be WWOX
Author Response
Dear Reviewer 1, thank you so much for your opinion about the paper. Please see the attachment for our responses. With kind regards, Żaneta Kałuzińska-Kołat.

Reviewer 2 Report
The authors devoted the article to determining the nature of WWOX in glioblastoma, since the influence of this gene on the malignant phenotype of GBM has already been reported.
The present study aimed to determine the main processes by which WWOX exhibits anti-cancer properties in GBM, taking into account the phenotypic heterogeneity between cell lines. Ectopic overexpression of WWOX was studied in T98G, DBTRG-05MG, U251MG and U87MG cell lines, which were compared using assays examining cell viability, proliferation, apoptosis, adhesion, clonogenicity, three-dimensional and attachment-independent growth, and invasiveness.
The observations demonstrating the antitumor properties of WWOX were the same for T98G, U251MG and U87MG. Increased tumor proliferation and growth was noted in DBTRG-05MG cells overexpressing WWOX.
The authors explained the observed effects using bioinformatics tools and showed that it was related to the transcription factor TARDBP and differences in USP25 and CPNE2 expression, which regulate the amount of EGFR surface. Thus, WWOX manifests its anticancer nature mainly by reducing cell viability and the invasiveness of glioblastoma.
I really liked the article. The design of the study is beyond doubt, the material is presented clearly and consistently, all observed patterns are explained. I believe that the article will undoubtedly interest readers.
Author Response
Dear Reviewer 2, thank you so much for your opinion about the paper. Please see the attachment for our responses. With kind regards, Żaneta Kałuzińska-Kołat.

Reviewer 3 Report
In an original study, Polish authors explain the role of WWOX in glioblastoma pathophysiology. Overall, the manuscript is carefully prepared.
Below I attach comments to the paper:
1. introduction - I have no critical remarks,
2. materials and methods - I have no critical comments here; only the authors may consider moving the chapter "statistical analysis" after "bioinformatics analysis,"
3. results - congratulations to the authors for the clear presentation of the results,
4. discussion:
a. please add limitations of the study,
b. please discuss the potential role of exosomes in the modulation of WWOX expression using these publications:
1. Exosomes containing differential expression of microRNA and mRNA in osteosarcoma that can predict response to chemotherapy,
2. Small Extracellular Vesicles and Their Involvement in Cancer Resistance: An Up-to-Date Review
Author Response
Dear Reviewer 3, thank you so much for your opinion about the paper. Please see the attachment for our responses. With kind regards, Żaneta Kałuzińska-Kołat.

Round 2
Reviewer 3 Report
The authors have satisfactorily responded to all my questions and made the necessary changes to the manuscript.Congratulations! I wish you all the best.